# Pathogenesis, Diagnosis, and Clinical Implications of Hereditary Hemochromatosis—The Cardiological Point of View

**DOI:** 10.3390/diagnostics11071279

**Published:** 2021-07-16

**Authors:** Ludmiła Daniłowicz-Szymanowicz, Michał Świątczak, Katarzyna Sikorska, Rafał R. Starzyński, Alicja Raczak, Paweł Lipiński

**Affiliations:** 1Department of Cardiology and Electrotherapy, Medical University of Gdańsk, Dębinki 7 St., 80-211 Gdańsk, Poland; michal_swiatczak@gumed.edu.pl; 2Department of Tropical Medicine and Epidemiology, Medical University of Gdańsk, Dębinki 7 St., 80-211 Gdańsk, Poland; katarzyna.sikorska@gumed.edu.pl; 3Department of Molecular Biology, Institute of Genetics and Animal Breeding, Polish Academy of Sciences, Wólka Kosowska, 05-552 Jastrzębiec, Poland; r.starzynski@ighz.pl (R.R.S.); p.lipinski@igbzpan.pl (P.L.); 4Clinical Psychology Department, Faculty of Health Sciences, Medical University of Gdańsk, 80-211 Gdańsk, Poland; alicja.raczak@gumed.edu.pl

**Keywords:** hereditary hemochromatosis, *HFE* gene, heart damage

## Abstract

Hereditary hemochromatosis (HH) is a genetic disease leading to excessive iron absorption, its accumulation, and oxidative stress induction causing different organ damage, including the heart. The process of cardiac involvement is slow and lasts for years. Cardiac pathology manifests as an impaired diastolic function and cardiac hypertrophy at first and as dilatative cardiomyopathy and heart failure with time. From the moment of heart failure appearance, the prognosis is poor. Therefore, it is crucial to prevent those lesions by upfront therapy at the preclinical phase of the disease. The most useful diagnostic tool for detecting cardiac involvement is echocardiography. However, during an early phase of the disease, when patients do not present severe abnormalities in serum iron parameters and severe symptoms of other organ involvement, heart damage may be overlooked due to the lack of evident signs of cardiac dysfunction. Considerable advancement in echocardiography, with particular attention to speckle tracking echocardiography, allows detecting discrete myocardial abnormalities and planning strategy for further clinical management before the occurrence of substantial heart damage. The review aims to present the current state of knowledge concerning cardiac involvement in HH. In addition, it could help cardiologists and other physicians in their everyday practice with HH patients.

## 1. Introduction

Hereditary hemochromatosis (HH) is one of the most common inherited metabolic diseases among Caucasians, occurring with a frequency of 1–2 for 500 people [1,2]. This heterogeneous group of iron overload disorders is caused by mutations in various genes encoding proteins involved in iron homeostasis [3,4,5,6,7]. More than 80% of HH cases are related to homozygosity for the *C282Y* mutation in the *HFE* gene. Other polymorphisms in the *HFE* gene are the *H63D* and *S65C* mutations. The *HFE* protein scheme is present in Figure 1.

Dysfunction of molecules that control iron homeostasis leads to excessive iron absorption in the duodenum and upper section of the small intestine and its maldistribution. As there is no regulatory mechanism for iron excretion from the human body, iron is deposited in many organs (liver, pancreas, other endocrine glands, skin, joints, heart) throughout the disease [2,8,9,10,11]. Bioactive iron ions produce oxidative stress that destroys involved tissues [2,12,13]. In HH, the intestinal absorption of iron is high, and multi-organ iron overload with organ failure may occur over decades. Due to the intense iron intake, cardiomyocytes are very susceptible to this type of damage [2,14]. 

Myocardial iron loading of the heart is well known as a possible complication in HH in diastolic and systolic functions [15,16,17,18,19]. Congestive heart failure is, among others, a late symptom of the disease, which develops after many years of its duration. It is accounted for about one third of deaths in natural hemochromatosis and cases of untreated HH [8,9,20] and was known as an essential factor that made qualification to liver transplantation impossible in patients with coexistent liver cirrhosis, liver failure, and hepatocellular carcinoma, all of which related to iron overload [1]. 

Genetic testing enables to detection of HH at an early stage and, consequently, to start treatment early enough to stop the alterations in different organs, including the heart [9,15,21,22,23,24]. However, when patients do not present severe abnormalities in serum iron parameters and severe symptoms of other organ involvement, the heart damage may be overlooked due to the lack of evident signs of cardiac dysfunction in routine cardiological screening.

The present review aims to systematize, based on the recent discoveries, the current state of knowledge concerning cardiac involvement in HH from a molecular mechanism of iron regulation to clinical presentation. The document could help cardiologists and other physicians’ practice with HH patients to plan the clinical observation and treatment model.

## 2. Methods

PubMed, Scopus, and Wiley databases were filtered for relevant publications regarding hereditary hemochromatosis. A search in electronic databases was conducted with free-text terms for pathogenesis, treatment, and diagnosis of hereditary hemochromatosis, cardiac hemochromatosis, cardiac involvement in hemochromatosis, and an iron overload disease. We looked at source materials from 1955–2020 and found 171 publications, including 91 original articles, 67 review papers, 9 case reports, and 4 brief communications 66 of them were related to cardiological issues. Eventually, 114 papers were selected for use in the article creation process, basing on the impact of the latter studies on current patient management. During the writing of the review, we additionally used guidelines directly related to liver and heart damage. 

In preparing the content of our manuscript, we followed the Preferred Reporting Items Requirements for Systematic Reviews and Meta-Analysis (PRISMA) guidelines whenever possible.

## 3. Results

### 3.1. Molecular Mechanisms of Iron Regulation

#### 3.1.1. Outline of Cellular Iron Metabolism

All aspects of intracellular iron homeostasis are mirrored in the so-called labile iron pool (LIP), a low-molecular-weight pool of weakly chelated iron in the cytosol that rapidly transits through the cell source of iron for heme synthesis and iron-sulfur cluster biogenesis. It has been claimed that LIP is also a cellular source of iron participating in the Fenton reaction and a sensor for the intracellular IRE/IRP regulatory mechanism. It seems that under physiological conditions, a LIP level is midway between the cellular need for iron and the hazard of excessive generation of hydroxyl radical (OH^•^) [25]. Iron concentration in LIP is primarily determined by the rate of iron uptake, storage, and release, accomplished mainly by three proteins, transferrin receptor 1 (TfR1), ferritin (Ft), and ferroportin (Fpn), respectively [26,27]. However, there are several exceptions to this general scheme of cellular iron handling, and its regulation depends on the cell type and function. Although most cells acquire plasma iron via TfR1-mediated endocytosis of transferrin (Tf)-bound iron [28], macrophages ingest large amounts of this microelement via erythrophagocytosis of senescent/damaged red blood cells [29]. On the other hand, absorptive enterocytes take up dietary inorganic iron from the gut mainly through the divalent metal transporter 1 (DMT1) [30]. 

Interestingly, growing evidence indicates that the molecular pattern of iron entry into and egress from cells may change under systemic iron overload [31] or hemolytic conditions [32]. Again, cardiomyocytes are the best illustration of this thesis.

#### 3.1.2. Molecular Mechanisms of Iron Overload in HH

*HFE* gene encodes *HFE* protein; a ubiquitously expressed atypical major histocompatibility class I-like molecule. *HFE* is a membrane protein that heterodimerizes with β2-microglobulin [33]. *C282Y* mutation disrupts a disulfide bond in the α3 domain of *HFE*; it abrogates its binding to β2-microglobulin and, consequently, prevents *HFE* presentation on the cell surface [34]. Although identification of *HFE* mutation(s) was a milestone in early diagnosis of hemochromatosis, at the time when the *C282Y* mutation has been identified (1996) [35], the mechanisms through which mutated *HFE* causes systemic iron overload and the role in the upregulation of intestinal absorption of dietary iron was unclear. In 2001 [36,37], the discovery of hepcidin shed light on the role of *HFE* in the sensing of systemic iron levels. Several studies have demonstrated that mutation in the *HFE* gene and some other genes involved in iron homeostasis leads to down-regulation of hepcidin (Hepc) expression, and in consequence, promotes intestinal iron absorption, resulting in systemic iron overload [38]. 

### 3.2. Mechanism of Cardiac Damage in HH

#### 3.2.1. Regulation of Iron Metabolism in Cardiomyocytes

Our current iron metabolism knowledge and regulation in cardiomyocytes derives from studies on mice with targeted cardiac-specific disruption of iron-related genes [39,40,41,42]. Under physiological conditions, diferric Tf that binds to the high-affinity TfR1 is used by cardiomyocytes as the main source of iron taken from the serum [43]. Mouse cardiomyocytes with inactivated *TfR1* gene become severely iron deficient, a phenomenon associated with heart failure and a strongly shortened lifespan [39]. These abnormalities could be partially reversed by aggressive iron therapy leading to the appearance of non-Tf bound iron (NTBI) and its uptake by cardiomyocytes. Indeed, NTBI is thought to play a major role as a source of iron for cells in various iron overload conditions. NTBI is a multicomponent pool including a considerable proportion of protein-bound iron [31]. Multiple routes of iron entry into cardiomyocytes exist under iron overload conditions. These include both L-type (LTCC) and T-type (TTCC) calcium channels [44], proteins from ZIP family and divalent metal transporter 1 (DMT1) [45]. The entry of iron via all these transporters seems to be beyond the control of canonical regulatory systems of cellular iron homeostasis and may potentially contribute to iron overload in the heart. There is substantial evidence that intracellular iron content can also be regulated through the Fpn-dependent pathway of iron release from cells to the extracellular environment [46]. Studies on mice with a cardiomyocyte-specific deletion of the *Fpn* gene showed a high increase in both iron and ferritin levels in cardiomyocytes [40]. This cardiomyocytic iron overload was associated with abnormal heart morphology, dilated cardiomyopathy, and reduced survival of *Fpn* knock-out mice [40]. Apart from being influenced by iron fluxes from and to cells, the LIP level is highly dependent on ferritin (Ft), an iron storage molecule with the ability to sequester up to 4500 iron atoms per molecule and thus playing the dual functions of iron detoxification and iron reserve. Ft forms a complex of 24 subunits consisting of a mixture of Ft heavy (H-Ft) and light (L-Ft) chains, showing different functional activities. Heart tissue contains primarily H-Ft-rich ferritins, which contrast to L-Ft-rich ferritins with relatively low iron content [47]. Importantly, H-Ft-rich ferritins turnover more rapidly and take and release iron more rapidly than L-Ft-rich ferritins [48]. This may be due to the process of ferritinophagy (autophagic degradation of ferritin), which is mediated by selective cargo-receptor Nuclear Receptor Coactivator-4 (NCOA4) that binds specifically H-Ft subunit and targets the whole protein to emerging autophagosome [49,50]. Considering accelerated catabolism of heart Ft, it is tempting to suggest that, especially under conditions of sustained and increased iron delivery, the potential of detoxifying iron is much less than in tissues with the main storage function containing up to 90% L-Ft subunits. This may be a reason for the largely reported susceptibility of cardiomyocytes to oxidative stress [51]. In this context, it tempting to note that the evidence of the manifestation of ferroptosis, a newly identified, dependent on iron and reactive oxygen species form of regulated cell death in iron overload cardiomyocytes, is emerging [52].

A cardiomyocyte-targeted deletion of *Irp1* and *Irp2* genes in mice resulted in a concerted up-regulation of Fpn and Ft and down-regulation of TfR1 in the heart. They led to the development of cardiac iron deficiency, associated with impaired mitochondrial respiration and cardiac energetics [42]. Interestingly, IRP-targeted mice were rescued by intravenous iron supplementation.

It is largely accepted that intracellular iron balance also depends on systemic regulation that relies on the interaction of hormone Hepc with Fpn [53]. The iron regulatory hormone Hepc, synthesized mainly in hepatocytes, limits iron fluxes to the bloodstream by promoting degradation of ferroportin Fpn in target cells, primarily macrophages, enterocytes, and hepatocytes. Thereby, Hepc decreases iron transfer into blood plasma from the duodenum, from macrophages involved in recycling senescent erythrocytes, and iron-storing hepatocytes. However, Hepc derived from hepatocytes may also induce iron sequestration within other Fpn-expressing cells, including cardiomyocytes, which are not essential for maintaining systemic iron homeostasis. Hepc expression has also been reported in many other mammalian cells, including cardiomyocytes [54]. For the first time, the biological role of tissue-specific Hepc has been demonstrated in mice with cardiomyocyte-specific deletion of the *Hamp* gene showing its crucial importance for the regulation of iron metabolism in the heart [41]. These mice showed cardiomyocyte iron deficiency (associated with fatal contractile and metabolic dysfunction) due to the upregulation of Fpn in cardiomyocytes caused by the absence of negative regulation via cardiac Hepc. Interestingly, circulating liver-derived Hepc did not compensate for the lack of peptide produced locally in the heart [41]. 

It is known that systemic Hepc insufficiency causes hyperabsorption of dietary iron, hyperferremia, and tissue iron overload, which are hallmarks of HH [33]. Impairment of the physiological mechanism of regulation leads to Hepc deficiency and, hence, inadequate with the need for increased iron absorption. In this way, primary hemochromatosis differs from secondary iron overload, which develops due to numerous blood transfusions in anemia, e.g., beta-thalassemia, spherocytosis, sickle cell anemia, myelodysplastic syndrome.

The heart is mentioned among organs most charged with iron in various animal models of hemochromatosis [55]. Precise analysis of iron distribution in cardiac tissue of mice with systemic deletion of *Hamp* gene (Hamp^-/-^) revealed unexpectedly that iron accumulates mainly in non-cardiomyocytic cells [40]. Although cardiac iron loading was much more significant in the heart of Hamp^-/-^ mice than in animals with cardiomyocyte-specific deletion of *Fpn* gene, only slightly elevated iron content was detected in cardiomyocytes of Hamp^-/-^ animals. This finding once again underlies the role of Fpn as a critical regulator of iron content in cardiomyocytes. A summary of the regulation of iron transport in cardiomyocytes is shown in Figure 2.

#### 3.2.2. Molecular Mechanisms of Cardiac Damage in HH

Although many studies have been devoted to the molecular aspect of HH, the mechanism of heart failure induced by iron excess is not known. Overwhelming serum transferrin capacity to bind iron observed in HH patients [31] leads to noncontrolled iron entry to cardiomyocytes (Figure 2) and, in turn, may increase cell susceptibility to oxidative stress. Oxidative stress in heart muscle decreases electromechanical coupling, inhibits SERCA2 enzyme function, and leads to the increase in cytoplasmic concentration of Ca ions in cardiomyocytes, leading to impaired relaxation and delayed contraction. Furthermore, oxidative stress induces peroxidation of cellular membranes, including mitochondrial membranes, resulting in a decreased ATP production in oxidative phosphorylation [56]. Besides, free iron ions may damage mitochondrial and nuclear DNA through the generation of oxidative stress and activates fibroblasts proliferation and differentiation to myofibroblasts responsible for heart fibrosis.

Interestingly, the number of mitochondria in cardiomyocytes is more significant than other cells, hence their greater vulnerability to iron overload, which manifests by the appearance of NTBI in HH patients. From the clinical and practical point of view, it is worth noticing that young adults with *HFE*-hemochromatosis may experience heart muscle damage when, among laboratory abnormalities, they present mainly high transferrin saturation values without an expected significant increase in the ferritin level. However, the level of NTBI is increased, leading to heart damage at the earliest stages of the disease. Significant hyperferritinemia occurs with the duration of the illness when effective treatment is not initiated.

#### 3.2.3. Cardiac Involvement in HH

Iron in the heart is mainly located inside the cardiomyocytes rather than in the extracellular matrix, suggesting that iron overload cardiomyopathy is storage rather than an infiltrative process [1]. The accumulation of iron deposits occurs starting from the epicardial, then through the myocardium to the endocardial layer [1]. Hypertrophy is one of the macroscopic consequences of the disease. On a mouse model, Sukumaran et al. [57] analyzed the mechanism of cardiac hypertrophy in HH. It consists of an increase in the expression of heavy myosin chains, with a shift in the number of chains α (decrease) and β (significant increase). The β-chains are responsible for the slower contraction of cardiomyocytes and are a marker of hypertrophy and, as a consequence, diastolic dysfunction. Dilated cardiomyopathy and heart failure are other cardiac complications of the HH, which are explained by different mechanisms. According to the first theory, the lack of *HFE* gene function directly impacts myocardial fibrosis [57]. The increased risk of coronary artery disease due to an increased iron level is the next cause [58]. According to another hypothesis, the *HFE* gene mutation (and the loss of *HFE* gene function) could directly affect the appearance of dilated cardiomyopathy [59]. The last hypothesis assumes the possibility of autoimmune process participation as a cause of heart failure in HH [60]. All this results in initial hypertrophy, diastolic dysfunction, heart dilatation, systolic dysfunction, and arrhythmias as possible clinical presentations of the HH. 

### 3.3. Clinical Presentation and Diagnostic of HH

Cirrhosis, diabetes, and dark skin color were the main symptoms of HH before the era of genetic testing revealing, whereas cardiac dysfunction was the essential clinical complication of this disease. The discovery of the *HFE* gene in 1996 and introduction to routine diagnostics of genetic tests in patients with abnormal iron management parameters indicating its excessive accumulation makes possible HH diagnosis at the early stages before the patients demonstrate symptoms of advanced diseases [61]. It is worth noting that iron is first deposited in the liver and later in the heart, but at the same time, as treatment continues, iron is removed from the liver much faster than from the heart [62]. Instead of the abovementioned classic triad of symptoms, nowadays, general malaise, arthralgia, hepatomegaly, and insensibly elevated transaminases are the main. Moreover, one of the early symptoms noticed by nowadays patients with HH is severe chronic fatigue, which substantially decreases the quality of life. Despite the significant intensification of fatigue and its frequent occurrence in HH, there are no studies in the literature on its nature and fatigue assessment method in these patients. Our experience shows that it could be essential in the clinical monitoring of HH patients in determining the frequency of phlebotomies, which we precisely described in the presentation of our clinical case presentation [63]. The level of fatigue in this patient was significantly elevated before the treatment (fatigue assessment scale: 25 points; Chalder fatigue scale: 12 points; fatigue severity scale: 37 points), whereas we observed a clear trend towards the decrease in the intensity of fatigue within venesections. It is interesting that due to loss iron during menstrual bleeding, premenopausal women’s progression of HH is slow; this is why symptomatic HH is more common in adult men than in adult women.

#### Laboratory Diagnostics

Patients with HH could have different abnormalities in laboratory tests. Hemoglobin and hematocrit are usually increased; in premenopausal women, it could be at the upper limit of the norm [64,65,66]. As liver involvement is widespread, increased levels of alanine transaminase and aspartate transaminase are possible to recognize; there are also frequent disturbances in serum lipids, most often in the form of primary hypertriglyceridemia [66,67,68]. The pancreas is another organ that can also be damaged in HH, leading to the development of impaired glucose tolerance or diabetes, manifested by elevated glucose levels in the human body [69].

In any HH clinical suspicion, measuring iron, ferritin, and transferrin saturation levels are recommended to be performed [2]. The following cut-off points are used for diagnosis of the pathological iron accumulation syndrome: ferritin >200 ng/mL for premenopausal women, >300 ng/mL for men and postmenopausal women, and transferrin saturation >45% [1]; these patients should be further assessed genetically. Total iron-binding capacity (TIBC) is also helpful as a parameter in diagnosing pathological iron accumulation syndromes, and its concentration is usually >450 mcg/dL or >80.55 mmol/L. The HH diagnostic algorithm is presented in Figure 3. 

It is worth noting that some of mentioned above laboratory abnormalities could be the consequences of other diseases. For instance, increased hematocrit level is a known abnormality in chronic obstructive pulmonary disease and pulmonary insufficiency [70]. Many hepatic pathologies are accompanied by increased transaminases levels [71,72]. As an acute-phase protein, ferritin may be elevated in inflammatory processes, including SARS-Cov-2 infection, cell necrosis, metabolic syndrome, alcohol abuse, and neoplastic diseases [1,73,74]. 

The essential laboratory parameter in HH assessment is NTBI level, which, due to the participation of these molecules in the generation of oxidative stress damaging the tissues, is particularly important [31]. High levels of NTBI may indicate earlier initiation of appropriate treatment to avoid damage to organs, including the heart muscle. Unfortunately, the routine determination of serum NTBI levels is impeded by a small number of laboratories with validated methods that allow obtaining irrefutable results [75]. In Europe, one of the most important centers that conduct a validated NTBI level assessment is the laboratory of prof. D.W. Swinkels in the Netherlands. 

### 3.4. Cardiological Diagnosis

#### 3.4.1. Electrocardiography

Electrocardiography is not a valuable and specific tool for HH diagnosis in asymptomatic or scantily symptomatic patients, how we can see in our group of the new-diagnosed HH (unpublished data). The ECG changes could be recognized in the late stages of the disease; however, they are not specific such as left ventricle hypertrophy, low-voltage QRS, or ST-T changes (as an expression of repolarization phase dysfunction). Iron overload of the heart conduction system can result in atrioventricular blocks due to the long-lasting disease [1]. Besides, supraventricular (mainly atrial fibrillation) and ventricular arrhythmias could be present in advanced HH stages. Interestingly, supraventricular arrhythmias correlate with oxidative stress levels but not with the level of iron overload, as demonstrated by Shizukuda et al. [76], whereas ventricular arrhythmias are not associated with any of the above factors. 

#### 3.4.2. Standard Echocardiography

Standard echocardiography is a well-known non-invasive method for revealing cardiac abnormalities. It should be considered one of the essential tools in diagnosing patients with HH in the late and early stages of developing that disease. This non-invasive, relatively cheap, and wide-available technique allows many HH patients to reveal cardiac abnormalities.

Cardiac hypertrophy and other features of diastolic dysfunction considered one of the earliest changes in long-lasting HH patients [15], are easy to detect by standard echocardiography. Tissue Doppler imaging allows revealing reduced early diastolic velocity (Em) of mitral annulus excursion (averaged for the values obtained from the lateral and septal corners of the mitral valve annulus). Shizukuda et al., additionally pay attention to the increased contractility of the left atrium as an early expression of diastolic dysfunction [77]. In many publications, changes reported in diastolic function in patients with HH were focused on patients with a long-lasting and long-treated HH [17,78,79]. We observed these changes in our patients with more than ten years HH history either. However, the group of patients with the newly diagnosed disease, showing no clinical symptoms yet, seems to be very interesting from a clinical point of view. In one of our previous studies [78], based on such patients, we found many differences in parameters referring to the diastolic function and the thickness of the left ventricle compared to healthy people. It is worth noting that differences were more expressed and included more parameters (Em, E/Em, left atrium area index, relative wall thickness) in older (≥50 years old) compared to age-matched healthy persons than in younger patients with HH. In our opinion, it can indirectly indicate a faster “aging” of the heart process in people with HH, but further research is needed on this issue. Noteworthy, all standard echocardiographic parameters measured in this study were normal; therefore, it would be impossible to reveal any diastolic dysfunction despite mentioned above differences. In this study, HH patients under 50 years of age did not differ in every mentioned above diastolic parameters, but those patients had significantly lower, but still within the normal range, left ventricular ejection fraction than the age-matched healthy group. We explained it by the more significant influence on systolic function in younger patients with newly diagnosed HH. Systolic dysfunction, which could be easily revealed by standard echocardiography, is considered to develop in later stages of the disease. Candell-Riera et al. point out increased dimensions of the heart cavities (late diastolic dimension of the left ventricle and size of the left atrium) with reduced shortening fraction and left ventricular ejection fraction [15] are typical for long-lasting and long-treated HH patients.

#### 3.4.3. Two-Dimensional Speckle Tracking Echocardiography (2D STE)

Two-dimensional (2D) STE is a novel technique of advanced echocardiography based on acoustic marker tracking methods, which seem more precise and distinctive than traditional echocardiography for detecting subtle myocardial abnormalities [80,81]. Data from literature described basal and apical LV rotation as an essential part of LV twist, which plays a crucial role in maintaining both systolic and diastolic function, whereas untwisting parameters were used to quantify LV diastolic function [82,83,84,85,86,87]. To date, only a few studies have employed speckle tracking analysis in the setting of iron-overload symptoms [88,89,90], but previously these results were based on patients with beta-thalassemia major, which is known as the extreme model of systemic iron overload [91]. The applicability of 2D STE for cardiac assessment in the early stages of hereditary HH, before substantial damage of the LV, seems to be of great clinical value [92,93]. We tried to precisely check it in one of our previous study [92] focused on patients with newly diagnosed HH. For the first time in the literature, we noticed that despite the absence of abnormalities of standard echocardiographic parameters, the patients were characterized by significantly lower values of apical and basal left ventricular rotation, as well as worse longitudinal strain values compared to healthy subjects (Table 1, Figure 4). 

Byrne et al. suggested that 2D STE guide intensification of venesection therapy showing the significant improvement of radial strains following a one-year course of venesection [93]. We presented similar changes in rotation, twist, and torsion LV parameters after six-month venesections in our patient with the early diagnosed HH [63]. We also noted no improvement in the global longitudinal strain in the observed patient, which can be associated with too short a follow-up period or irreversible changes in the myocardium. Based on our previous research, we could postulate that 2D STE seems to be the test of choice in patients with HH for detecting and monitoring early changes in the heart and for the implementation of treatment at the preclinical stage of the disease. It is noteworthy that, according to data from the literature and our research, correlations between 2D STE myocardial parameters and the iron indices in the serum are rather wicking or even absent [92,94]. This agrees with the results regarding other iron-overload syndromes [95,96], which suggest the absence of a “direct” relationship between routinely measured in the everyday clinical practice iron turnover parameters and myocardial damage. Myocardial iron overload seems not to be the only mechanism involved in the development of HH cardiomyopathy. On the other hand, as we presented in explaining the molecular mechanisms of cardiac involvement in HH, NTBI could have a significant impact on this issue.

#### 3.4.4. Three-Dimensional (3D) Real-Time Echocardiography

Three-dimensional (3D) real-time echocardiography is a novel technique for comprehensively evaluating the heart, including atria and ventricles volume, ejection fraction, and left ventricular mass [97]. This technique is considered to be comparable with magnetic resonance about the abovementioned parameters. Furthermore, some authors suggested 3D echocardiography to be helpful for earlier detection of left atrial dysfunction in asymptomatic patients with cardiac iron overload [98]. In one of our previous studies [94], we used 3D echocardiography in the evaluation of left ventricle volume and mass, as well as left ventricular ejection fraction in patients with early-diagnosed in comparison to long-lasting HH, and suggested that we may deal with hemochromatosis-induced cardiomyopathy, rather than with simple LV thickening. Therefore, 3D echocardiography seems to be an essential technique for HH patients’ assessment in the future.

#### 3.4.5. Cardiac Magnetic Resonance (CMR)

CMR is the principal imaging modality for the non-invasive assessment of myocardium in different pathology [99] CMR is a sensitive and specific tool in the diagnosis of cardiac abnormalities due to HH [100,101,102]. A reduction of T2 relaxation <20 ms is considering a symptom of iron overload [91]. Carpenter et al. [103], in the paper devoted to the study concerning left ventricular dysfunction in HH using CMR noticed a correlation between the presence of cardiac siderosis (reduction of T2 relaxation time) and ferritin concentration >1000 ng/mL and, therefore, propose to perform CMR in each patient diagnosed with HH with ferritin >1000 ng/mL. The authors also point out that reducing T2 relaxation time is characteristic of iron overload, which allows differentiation of the cause of heart failure in patients with HH. In the abovementioned article, attention was paid to improving EF and extending the T2 relaxation time after several years of treatment with phlebotomy, suggesting that CMR is a valuable tool for monitoring therapy. Despite the indisputable value of CMR, the use of this modality across Europe is diverse, and CMR is not a first-line diagnostic technique in everyday clinical practice. This situation is multifactorial, and the limited access, economic issues, and deficits in training play a central role. Therefore, searching for new non-invasive diagnostic techniques like STE or 3D echocardiography seems crucial for determining those with strong indications for CMR.

#### 3.4.6. Biopsy

Biopsy of the heart, contrary to a liver biopsy, an essential diagnostic method of liver injury in HH, is not routinely performed to evaluate cardiac abnormalities. It has been established that myocardial siderosis is a patchy process-iron clusters can be seen among healthy tissue, that is why the risk of false-negative results of that test is considerable [104,105]. A cardiac biopsy should not be used in the diagnosis of asymptomatic patients; it is an invasive and risky examination, which, in the era of advanced echocardiography and magnetic resonance techniques, is limited to unclear situations that require distinguish from other causes of heart failure or infiltrative diseases [1].

### 3.5. Treatment of HH and ITS Influence on the Cardiac Function

For many decades phlebotomy has been used as the only method with successful results [1,2,106]. Unfortunately, some additional clinical problems, such as anemia, or hemodynamic instability, could be the essential contraindications to this type of treatment. In these situations, the alternatives are treated with chelating drugs [2,107], which are not widely used due to side effects, the high cost, and limitations in registered indications [108]. Phlebotomies are more effective in reducing iron levels than chelating drugs [109]. Erythrocytapheresis (selective red blood cell apheresis) is another therapeutic option, slightly more effective than phlebotomy [110]; however, it is not used routinely due to the costs. Non-pharmacological treatment, like a diet with avoiding iron-fortified meals, raw shellfish, vitamin C-rich products, and alcohol consumption, significantly impacts reducing iron levels. 

The appropriate treatment proved to reverse cardiac damages [15,21,63,91,103,110,111]. Candell-Riera et al. showed a significant improvement in left atrial size, diameters, fractional shortening, ejection fraction, and LV mass with venesections [15]. Rombout-Sestrienkova et al. presented similar changes due to erythrocytapheresis treatment of a HH patient with a left ventricle assist device implanted due to a nonischemic dilated cardiomyopathy with severe systolic dysfunction [110]. Some publications concerning the positive effect of chelating drugs on the left and right ventricles parameters [112,113]. There are some crucial data in the literature concerning the use of STE in monitoring the effects of therapy in patients with HH. Byrne et al. in their paper, paid attention to the improvement of the radial strain, isovolumetric relaxation time, and left the atrial force after a year of venesection treatment in patients with newly diagnosed HH [91]. Our experiences [63] also show that phlebotomy therapy leads to significant and fast improvement in rotation parameters, including twist and torsion of the LV. Based on our results, we note that the LV longitudinal strain could be the parameter, which may indicate irreversible damage of the heart or improve within longer than several months of treatment [63]. Some authors point out that T2 relaxation time measured in CMR also changes during HH treatment [21,103,111]. Carpenter et al. demonstrated an improvement in this parameter in patients with newly diagnosed HH treated with venesections [103], Tauchenová et al., showed it as a consequence of erythrocytapharesis [21], also Pennell et al., and Tanner et al. prove that after using of iron chelators [111,114]. 

## 4. Summary

*HFE* gene-dependent HH is a slowly developing genetic disease connected with HFE gene mutation and based on complex iron dysregulation mechanisms. Cardiac involvement is inevitable in long-lasting conditions, leading to cardiomyopathy—one of the leading causes of mortality in the group of untreated HH patients. Appropriate therapy was proved to reverse cardiac damages, and implementation of the therapy at the early stages of the disease is the main rule in HH patients. Therefore, apart from genetic testing and iron parameters measurement, the possibility of early detection of cardiac involvement using modern imaging techniques, even in asymptomatic patients, seems to be of great clinical value. However, further studies are needed to confirm whether these techniques would be helpful not only for diagnosis but also for regular monitoring of treatment.

## Figures and Tables

**Figure 1 diagnostics-11-01279-f001:**
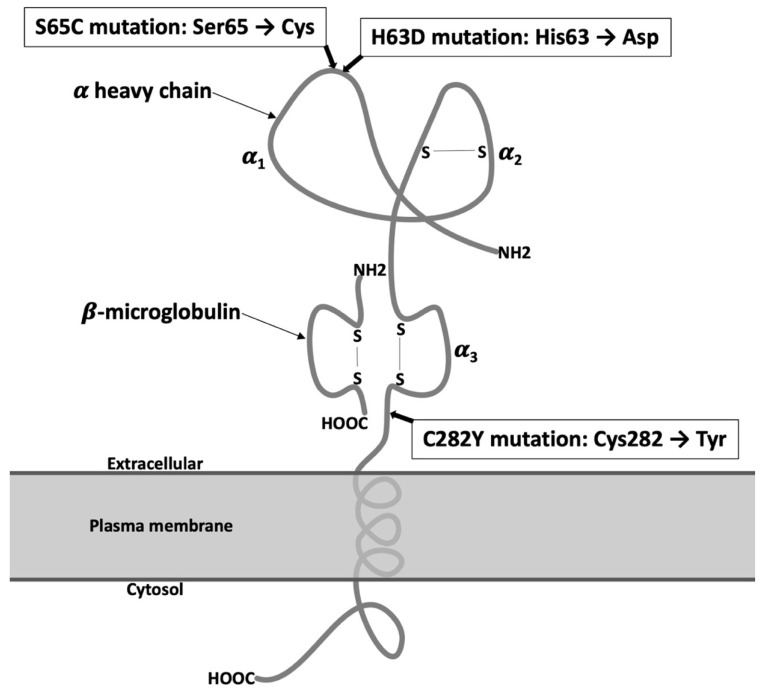
The *HFE* protein scheme presented with β2-microglobulin. This complex is located on the surface of the cell. The *HFE* protein consists of three domains: α1, α2, and α3. β2-microglobulin binds to a protein through an α3 domain. Positions of the three *HFE* mutations, *C282Y*, *H63D*, and *S65C*, are shown.

**Figure 2 diagnostics-11-01279-f002:**
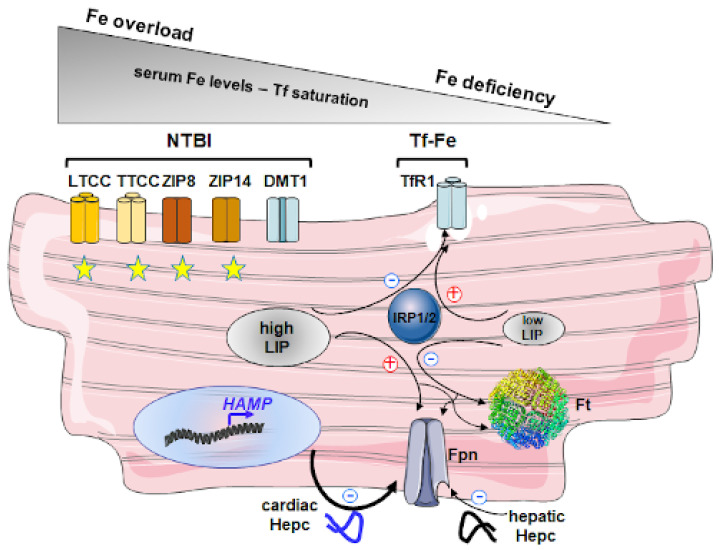
Regulation of iron import and export in cardiomyocytes under iron overload and iron-deficient conditions. In normal and iron-deficient conditions cardiomyocytes acquire iron via transferrin receptor 1 (TfR1)-mediated endocytosis of transferrin-bound iron (Tf-Fe), which is the main source of iron in the blood plasma. At a low iron concentration in the labile iron pool (LIP), direct interactions between IRPs and several IRE motifs stabilize TfR1 mRNA. The converse regulation of TfR1 synthesis, being a consequence of the lack of binding of IRPs to IRE, occurs in cells with high iron in LIP. Iron systemic overload is characterized by both high plasma iron concentration and transferrin saturation and the appearance of non-transferrin bound iron (NTBI). Under these conditions, new routes of iron transport into cardiomyocytes are an option including both L-type (LTCC) and T-type (TTCC) calcium channels, proteins from ZIP family such as ZIP8 and ZIP14. The expression of these importers remains beyond the control by IRP/IRE posttranscriptional mechanism and thus they may potentially contribute to heart overload. Regarding DMT1, only the iron-responsive element (IRE)-containing form, which corresponds to one of two splice forms of DMT1, is responsive to intracellular iron level iron via IRPs and should be down-regulated when LIP is elevated. However, it is not clear whether this form is abundant in cardiomyocytes. Iron in excess of metabolic needs is stored in a soluble and non-toxic form inside ferritin (Ft). In iron-deficient cardiomyocytes, the binding of IRPs to the unique IRE in the 5′-UTR of Ft subunits mRNAs blocks the translation initiation. The converse regulation of Ft occurs in iron-replete cells. Iron export from cardiomyocytes largely depends on ferroportin (Fpn), sole identified cell iron exporter. Its expression in response to iron is regulated intra- and extracellularly. Intracellular regulation involving the IRP/IRE system is the same as that of Ft. Extracellular regulation is mediated by hepcidin (Hepc), a 25-aa peptide predominantly produced by the hepatocytes in response to systemic iron availability. Hepc is also found in heart cells. However, in contrast to hepatic Hepc, iron-dependent mechanisms of the regulation of cardiac peptide are not known. Hepcidin inhibits cellular efflux of iron by binding to Fpn, subsequent internalization, and degradation of Fpn in lysosomes. It has been proposed that retention of iron in cardiomyocytes is mostly due to autocrine regulation by cardiac Hepc, which outcompetes paracrine hepatic Hepc dependent one.

**Figure 3 diagnostics-11-01279-f003:**
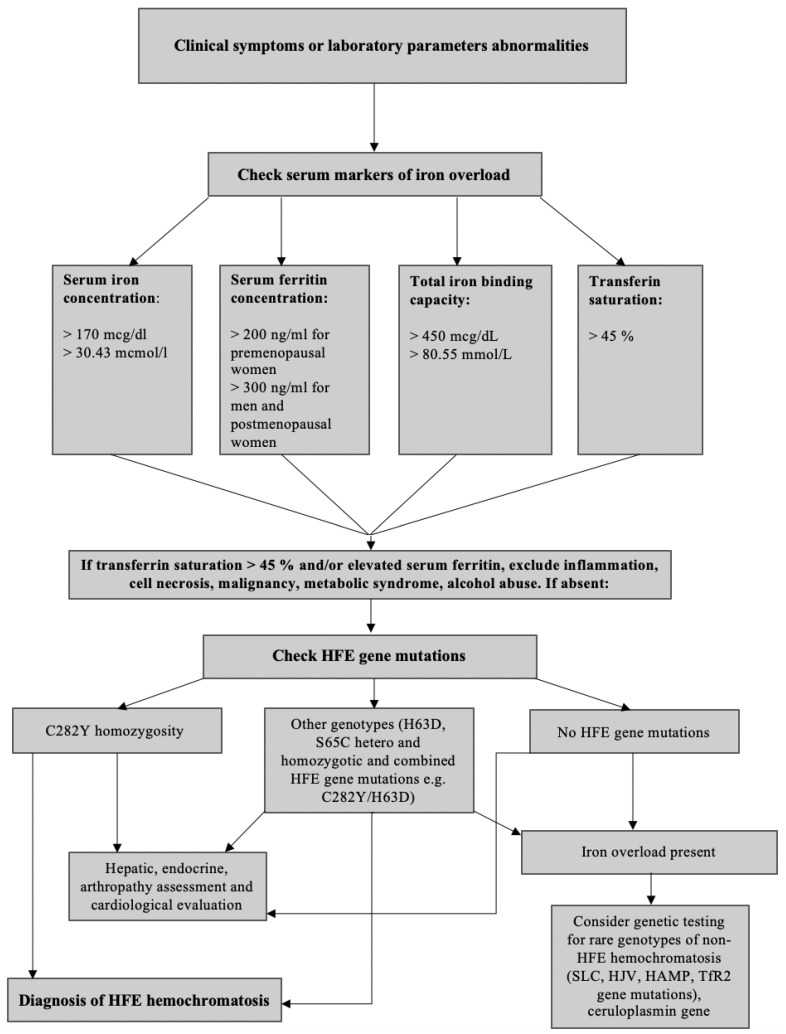
Diagnostic algorithm for suspected hemochromatosis.

**Figure 4 diagnostics-11-01279-f004:**
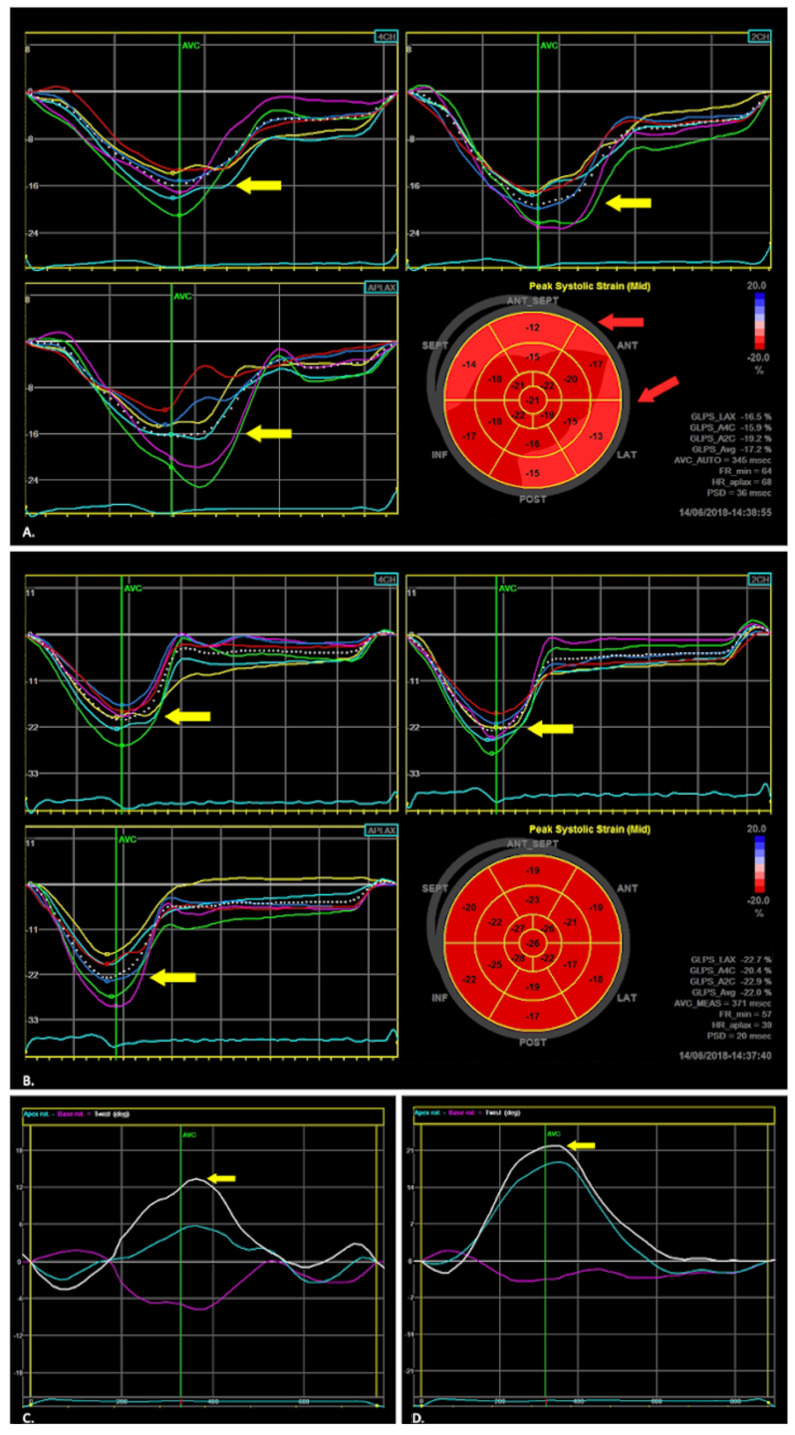
Examples of 2D STE measurements in HH patients and healthy volunteers. Global LV longitudinal strain measurements show slightly reduced longitudinal regional myocardial deformation in long-axis, 2- and 4-chamber views (yellow arrows) in the HH patient (**A**) in comparison to healthy volunteer (**B**), with more prominent impairment at the basal segments (red arrows), without the involvement of cardiac apex in the “bull’s-eye” map. The examples of rotation parameters (**C**,**D**) show LV apical (blue line) and basal (pink line) rotation curves, LV twist curve (white line), and peak LV twist (arrow) [92].

**Table 1 diagnostics-11-01279-t001:** Methods for cardiac abnormalities assessment in HH.

Cardiac Diagnostic Comparison		
Method	Characteristics	
Electrocardiography	Possible features of hypertrophy and nonspecific ST-T changes in the long-lasting HH	
Echocardiography	Standard echocardiography (abnormalities possible to be discovered in the long-lasting HH)	LV hypertrophy LA enlargement Diastolic LV dysfunction Systolic LV dysfunction
	Two-dimensional speckle tracking echocardiography (abnormalities possible to be discovered at the early stages of HH)	Lower values of apical LV rotation Lower values of basal LV rotation Lower global longitudinal strain of LV
	Three-dimensional echocardiography * abnormalities possible to be discovered at the early stages of HH ** abnormalities possible to be discovered in the long-lasting HH	Higher LV thickness (*/**) Higher LV mass (*/**) Higher LV long axis length (*/**) Higher LA diameter and volume (*/**) Lower LV ejection fraction (*)
Cardiac magnetic resonance	Reduced T2 relaxation time	
Biopsy	Iron clusters among healthy tissue (*) Limitations: high risk of false-negative results	
	* limited to unclear situations that require distinguish from other causes of heart failure or infiltrative diseases	

HH—hereditary hemochromatosis; LV—left ventricle; LA—left atrium.

## Data Availability

Please refer to the corresponding author. Data depicted in Figure 4 are available with echocardiography (https://onlinelibrary.wiley.com/journal/15408175, accessed on 12 July 2021) permission for re-use by authors in another publication (https://onlinelibrary.wiley.com/doi/abs/10.1111/echo.14141, accessed on 12 July 2021).

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
