# Peer review of "Pathogenesis, Diagnosis, and Clinical Implications of Hereditary Hemochromatosis—The Cardiological Point of View"

_diagnostics, 2021, doi:10.3390/diagnostics11071279_

Round 1

Reviewer 1 Report

This is a review article about hereditary hemochromatosis (HH) manifesting in cardiology, and they suggested that uses of echocardiography could help.

However, I suggest the authors to refer the guide: BMJ 2019; 364 doi: https://doi.org/10.1136/bmj.k4597

Otherwise, The PRISMA 2020 statement is helpful very much.

Author Response

Dear Reviewer,

as the authors of the article, we are very grateful for the reliable review and valuable comment on the content of the paper. We would also like to thank you for the valuable time you devoted to reviewing our article.

We have tried to follow all the comments of Reviewer, trying to increase the value of our paper. We hope that we have met The Reviewer’s expectations regarding the revision of our article.

We would like to refer to the Reviewer’s specific comments:

  1. However, I suggest the authors to refer the guide: BMJ 2019; 364 doi: https://doi.org/10.1136/bmj.k4597. Otherwise, The PRISMA 2020 statement is helpful very much.

Thank you very much for this important suggestion. In preparing the content of our manuscript, we tried to follow the Preferred Reporting Items Requirements for Systematic Reviews and Meta-Analysis (PRISMA) whenever possible. After the Reviewer’s suggestion, we added the necessary "Methods" paragraph in the article (page 3, lines 117 to 131).

PubMed, Scopus, and Wiley databases were filtered for relevant publications regarding hereditary hemochromatosis. A search in electronic databases was conducted with free-text terms for pathogenesis, treatment, and diagnosis of hereditary hemochromatosis, cardiac hemochromatosis, cardiac involvement in hemochromatosis, and an iron overload disease. We looked at source materials from 1955 – 2020 and found 171 publications, including 91 original articles, 67 review papers, 9 case reports, and 4 brief communications 66 of them were related to cardiological issues. Eventually, 114 papers were selected for use in the article creation process, basing on the impact of the latter studies on current patient management. During the writing of the review, we additionally used guidelines directly related to liver and heart damage. In preparing the content of our manuscript, we followed the Preferred Reporting Items Requirements for Systematic Reviews and Meta-Analysis (PRISMA) guidelines whenever possible.

Sincerely yours,

Authors.

Reviewer 2 Report

The manuscript by Ludmiła and colleagues reviewed the current state of knowledge concerning cardiac involvement in hereditary hemochromatosis (HH). The authors summarized the molecular mechanisms of iron regulation and cardiac damage in HH. The clinical presentation and diagnostic of HH and cardiological diagnosis and treatment of HH and its influence on the cardiac function were also mentioned. Overall, this review is an interesting and my specific comments are listed below.

  1. In page 11, I think “Figure 2” should be “Figure 3”. And the figure legend is also confusing. Please revise this part.
  2. It would be better to show HFE mutations in the HFE protein diagram.
  3. The title “Hereditary hemochromatosis – what cardiologists need to know?” can’t summarize the content of this review article.
  4. For a review paper, it is better to make the comment combining with your own research.

Author Response

Dear Reviewer,

As the authors of the article, we are very grateful for the reliable review and valuable comments on the paper's content, which allowed us to increase its scientific value. We would also like to thank you for the valuable time you devoted to reviewing our article. We have tried to follow all of the Reviewer's comments, trying to increase the value of our paper. We hope that we have met the Reviewer’s expectations regarding the revision of our article.

We want to refer to your specific comments:

  1. "In page 11, I think “Figure 2” should be “Figure 3”. And the figure legend is also confusing. Please revise this part."

We thank the Reviewer very much for pointing out this oversight. Our mistake results from an overlooking in the preparation of the paper's content according to the journal's guidelines. This failure has already been corrected (page 12-13, lines: 500-507).

  1. "It would be better to show HFE mutations in the HFE protein diagram."

Thanks to the Reviewer for this valuable advice; we followed it up. We included a new figure 1 in the article that consists of the HFE protein scheme with marked positions of the three HFE mutations: C282Y, H63D, and S65C (page 1, lines: 38-40; Figure 1 – page 2, description of this figure – page 2, lines 47-50).

  1. The title “Hereditary hemochromatosis – what cardiologists need to know?” can’t summarize the content of this review article.

We thank the Reviewer very much for this important suggestion. We tried to change the title of our paper, trying to summarize better the content of our review (New title: “Pathogenesis, diagnosis, and clinical implications of hereditary hemochromatosis - the cardiological point of view”; page 1, lines: 2-3).

  1. For a review paper, it is better to make the comment combining with your own research.

Thank you very much for this remark. We have tried to expand the scope of information in the text of the article regarding our own research. The authors hope this answer will be acceptable for the Reviewer (page: 8, lines: 308-316; page 10, lines: 368-370, and 392-402; page: 11, lines: 478-483; page: 13, lines: 510-518, and 531-535; page: 15, lines: 649-654).

  1. We tried to check the English language and style and correct all the mistakes and hypos using a qualified English-speaking person. We are ready for further actions in this area whenever necessary.

Sincerely yours,

Authors.

Round 2

Reviewer 1 Report

I have no more comment.